# Digital Fitness Revolution: User Perspectives on Fitbit’s Role in Health Management

**DOI:** 10.3390/bs15020231

**Published:** 2025-02-18

**Authors:** Seong-bin Jang, Minseong Kim

**Affiliations:** 1Department of Hotel Airline Service & Tourism, Jeonnam State University, 152 Juknokwon-ro, Damyang-eup, Damyang-gun 57337, Republic of Korea; seongbin8964@naver.com; 2Department of Marketing and Information Systems, College of Business, Louisiana State University Shreveport, Shreveport, LA 71115, USA

**Keywords:** fitness technologies, user experience, Fitbit, physical activity motivation, user feedback analysis, digital health applications

## Abstract

This research explores the intersection of health informatics and behavioral science through the lens of fitness technologies, specifically Fitbit products. Grounded in the Technology Acceptance Model (TAM) and Self-Determination Theory (SDT), this study examines how these technologies influence user acceptance and physical activity motivation. Employing a qualitative approach, the paper analyzed Fitbit user reviews to reveal insights into real-world interactions and perceptions, thereby deepening the understanding of technology adoption behaviors in health contexts. The findings highlight the significance of perceived ease of use and usefulness, as well as the integration of health consciousness in technology acceptance, enriching the TAM framework. Additionally, the study confirms Self-Determination Theory’s proposition of intrinsic motivation being more effective for lasting behavior change, as seen in users’ evolving interactions with Fitbit features. Furthermore, this study contributes to health behavior theories by demonstrating the role of technological devices in altering health routines.

## 1. Introduction

The emergence of fitness technologies, notably mobile applications and smartwatches compatible with health monitoring, has revolutionized the way individuals engage with their health and wellness routines ([9]; [14]; [31]). Fitbit, as a prime example of this technological innovation, exemplifies the profound impact these devices have on encouraging and tracking physical activity, thereby significantly influencing both physical and mental well-being ([14]; [16]). This transformation is not limited to mere activity tracking; these technologies have expanded their reach to encompass a wide array of bodily functions and conditions ([9]; [33]). For instance, individuals can now continuously monitor their heart rate, sleep patterns, and even stress levels, providing a comprehensive view of their overall health status ([16]; [15]). This constant monitoring has immense implications for preventive healthcare, allowing for early detection of potential health issues and facilitating timely medical intervention ([9]). Moreover, these devices have become crucial in personalized health management ([33]; [31]). By analyzing data collected over time, individuals can gain insights into their unique health patterns and make informed decisions about diet, exercise, and lifestyle changes. This personalized approach is instrumental in fostering a proactive attitude towards health, shifting the focus from treatment to prevention ([14]; [33]). To respond to this trend, our research aims to explore the theoretical implications and practical contributions of these technologies, with a particular focus on the user experiences and perceptions of Fitbit products, underscoring their integral role in modern health and wellness practices.

In this academic research, the theoretical foundations are built upon several key frameworks in health informatics and behavioral science ([6]; [16]; [25]; [28]; [30]). At its core is the Technology Acceptance Model (TAM), which provides a framework for understanding user acceptance and engagement with health technologies ([6]; [25]). Our study also examines the perceived ease of use and usefulness of fitness apps and smartwatches, thereby contributing to a more nuanced understanding of technology adoption behaviors, particularly in health contexts. Additionally, the Self-Determination Theory is employed as a lens to investigate how these technologies can motivate and sustain physical activities, which are crucial for mental and physical well-being ([28]; [30]).

TAM serves as the foundational framework for this study due to its theoretical parsimony, empirical validation, and domain relevance in health and fitness technology adoption ([1]; [5]; [7]). For example, while UTAUT2 extends the original Unified Theory of Acceptance and Use of Technology (UTAUT) by incorporating additional constructs, such as hedonic motivation, price value, and habit, these factors are less central to the adoption of fitness wearables, where user decisions are primarily shaped by functional utility (usefulness) and usability (ease of use) rather than external incentives ([7]). Fitness tracking technologies, such as Fitbit, are often adopted in self-regulated, health-conscious contexts, where perceived usefulness (e.g., effective health monitoring) and perceived ease of use (e.g., intuitive interface) remain the most influential predictors of acceptance and sustained engagement ([1]). Additionally, TAM has been widely empirically validated in health technology studies, demonstrating high explanatory power in predicting adoption and long-term usage behaviors ([5]). Given its parsimony and theoretical rigor, TAM provides a more focused and adaptable model for assessing user attitudes toward Fitbit, particularly in relation to its role in health monitoring and behavioral reinforcement ([1]).

Consequently, our research makes a significant contribution to the existing literature on fitness mobile application acceptance behaviors, offering empirical insights into user experiences with Fitbit products. While many studies have examined the functionalities of health technologies in controlled environments, such as surveys or experiments, our study uniquely leverages user-generated reviews to extract insights into real-world user interactions and perceptions ([6]; [16]; [25]; [28]; [30]). This approach bridges the gap between theoretical models of technology acceptance and the actual experiences of users, providing a more comprehensive understanding of the role of fitness technologies in everyday health practices ([31]). From a practical standpoint, this research offers direct implications for the design and development of user-centric health technologies. By analyzing consumer reviews, this study identifies key factors that influence user satisfaction and engagement, which are crucial for the design of effective and appealing fitness applications ([15]). These insights are invaluable for developers and designers in creating fitness technologies that not only meet functional requirements but also resonate with users on a personal level. In addition, employing a qualitative analysis, this research provides a comprehensive overview of user experiences with Fitbit products. By bridging theoretical models and user experiences, our research provides a comprehensive view of the role of fitness technologies in enhancing physical activity and, consequently, mental and physical health ([9]; [14]).

This study begins with a detailed analysis of consumer reviews, followed by a discussion that situates these findings within the broader theoretical frameworks of health technology and user behavior. The conclusion outlines the implications for the design and functionality of fitness apps, contributing to the ongoing development of health technologies.

## 2. Literature Review

The burgeoning field of fitness technology, particularly wearable devices and mobile applications, has attracted significant academic interest in recent years ([9]; [15]; [25]; [28]; [33]). Research in this domain spans several key areas, each contributing to a nuanced understanding of how these technologies intersect with health and fitness behaviors ([9]). This literature review highlights the multi-faceted nature of research in fitness technology, underscoring the need for more nuanced, long-term, and culturally diverse studies. Addressing the identified academic gaps is essential for a holistic understanding of fitness technology’s role in health and wellness and for guiding the development of more effective and inclusive solutions in this rapidly evolving field ([30]).

Technology adoption has been widely studied through various theoretical lenses, each offering distinct perspectives on the factors that drive user acceptance and sustained engagement with digital innovations. Among the most influential models, UTAUT and its extended version, UTAUT2, have been widely applied to explain technology adoption in consumer settings ([7]). UTAUT2 introduces additional constructs, which are more pertinent to general commercial and entertainment-oriented applications, where external incentives and habitual behaviors play a larger role in adoption decisions. In contrast, fitness wearables operate within a self-motivated, health-conscious domain, where the key adoption drivers revolve around perceived usefulness and perceived ease of use—the core dimensions of TAM ([5]).

Diffusion of Innovation (DOI) theory offers another perspective, emphasizing how relative advantage, compatibility, complexity, trialability, and observability impact adoption ([18]; [23]). While DOI provides valuable insights into the diffusion of fitness technologies at a societal level, it does not explicitly focus on individual psychological determinants of acceptance, making it less suitable for analyzing user-level adoption behavior in fitness tracking technologies ([18]; [23]). Similarly, the Theory of Planned Behavior (TPB) highlights the role of attitude, subjective norms, and perceived behavioral control in shaping behavioral intentions. However, TPB does not fully capture the role of system design, usability, and perceived technological benefits ([17]), which are central to understanding fitness wearable adoption.

Extensive studies applying TAM and Self-Determination Theory have explored factors driving the adoption of fitness technologies ([6]; [16]; [25]; [28]; [30]). These studies underscore the role of usability, functionality, and personal health goals in influencing user attitudes and behaviors towards technology ([6]; [14]; [25]). However, a noticeable gap exists in empirical research examining the evolution of these attitudes and behaviors within strictly controlled environments, particularly when aligned with the methodologies favored by scholars in this field ([6]; [16]; [28]). Research assessing the effectiveness of wearable fitness devices in promoting physical activity often yields results that are expected or hypothesized, mirroring empirical findings observed in other technological contexts ([15]). Additionally, some previous studies have acknowledged their limitations, specifically the lack of consideration for health-related variables in their expanded models or theories ([9]). A critical gap in this area is the absence of impartial studies that explore both the health and technological impacts, as well as the comprehensive integration of these devices into users’ overall health management through physical activities ([30]).

The role of health awareness in technology adoption has gained significant attention in the field of health informatics and behavioral science. Wearable fitness technologies, such as Fitbit, serve not only as tracking devices but also as informational tools that enhance users’ awareness of their health metrics ([15]). Health awareness is defined as an individual’s ability to monitor, interpret, and respond to biometric feedback, which, in turn, influences behavioral decisions and lifestyle modifications ([4]). Prior research indicates that individuals who actively track health indicators such as heart rate, sleep patterns, step count, and calorie expenditure develop a greater sense of control over their well-being ([21]). This suggests that individuals are more likely to adopt health technologies when they perceive that these tools provide real-time, actionable insights that allow for immediate behavioral adjustments ([14]). Our analysis of Fitbit user reviews demonstrates that continuous engagement with biometric tracking enhances health awareness. Users frequently describe how Fitbit helps them recognize and modify their physical activity, sleep habits, and overall health choices (e.g., “I never realized how little I moved until I got my Fitbit. Now, I make sure to walk at least 10,000 steps daily”). Similarly, another user shared how Fitbit’s heart rate tracking feature provided critical insights (e.g., “I never paid attention to my heart rate before, but after using Fitbit, I noticed patterns in my heart rate changes and started adjusting my workouts accordingly”). These testimonials illustrate how Fitbit’s real-time feedback loop encourages health-conscious behavior changes, reinforcing the role of health awareness as an adoption driver.

While TAM emphasizes perceived usefulness and ease of use as key drivers of technology adoption, emerging research suggests that health awareness should be considered a critical extension of TAM in health technology adoption ([6]). Wearable fitness technologies differ from general-purpose digital devices in that they actively engage users in continuous self-monitoring, reinforcing perceived usefulness through personalized health feedback ([14]; [33]). Users who develop an increased awareness of their physical activity levels, cardiovascular health, and sleep quality are more likely to perceive these devices as essential health management tools rather than simple fitness accessories. Given these insights, future extensions of TAM in health technology research should account for health awareness as a determinant of adoption and sustained engagement. While perceived ease of use and usefulness remain crucial, the ability of wearable devices to enhance users’ self-regulation and decision-making capabilities plays an equally important role in their long-term utility ([28]).

Investigations into the roles of gamification, social features, and feedback in fitness apps have underscored their significance in maintaining user engagement and adherence to fitness routines, as evidenced by studies from [3] ([3]), [11] ([11]), [19] ([19]), and [26] ([26]). However, there is a pressing need for more comprehensive exploration into the psychological processes driving these behavioral changes, especially in terms of users’ satisfaction with the technology, a perspective highlighted by [15] ([15]). The critical role of gamified functional aspects in fitness apps, such as customizable goals and interfaces, in enhancing user satisfaction has been established by [3] ([3]) and [19] ([19]). Nevertheless, research focusing on adaptive personalization—where the technology dynamically adjusts to the evolving needs and behaviors of users in health or physical activity contexts—remains limited, as indicated by [11] ([11]) and [19] ([19]).

The increasing collection and use of personal health data by fitness technologies have heightened concerns about privacy and security. While studies like those by [9] ([9]) and [32] ([32]) have started to explore user attitudes towards data sharing and trust in technology providers, there remains an academic gap in understanding user perspectives within the health- and physical-activity-focused contexts, especially in light of evolving data privacy regulations. Furthermore, research by [9] ([9]), [12] ([12]), and [32] ([32]) has examined how cultural and demographic factors influence the adoption and use of fitness technology. Prior research in this field reveals varying patterns of technology adoption and usage across different groups, underscoring the need for more comprehensive and unbiased research to address these differences ([14]).

In this study, the comprehensive review of literature highlights a significant academic gap in the field of health and fitness technology. Notably, the lack of uncontrolled studies restricts our understanding of how real-world user experiences with fitness technologies contribute to their satisfaction, especially as users become more familiar and loyal to these devices and technologies. Moreover, there is a shortfall in comprehensive research on adaptive personalization in fitness apps, which could lead to more tailored and effective user experiences. Additionally, the field lacks extensive cross-cultural studies that investigate the impact of cultural norms and demographic variations on technology design and effectiveness. Lastly, the implications of evolving privacy laws on user trust and attitudes towards fitness technologies represent a promising area for future exploration.

## 3. Methods

### 3.1. Study Design

Fitbit was selected as the primary focus of this research due to its prominence in the health and fitness technology market, as well as its long-standing reputation as an industry leader in wearable health-tracking devices ([29]). Since its inception in 2007, Fitbit has continuously evolved to offer a comprehensive suite of features, including heart rate monitoring, sleep tracking, activity tracking, calorie counting, and integration with various third-party health applications. These features make Fitbit one of the most widely adopted wearable brands among fitness enthusiasts, health-conscious individuals, and users seeking personalized health insights. According to market research reports, Fitbit holds a significant share of the global wearable fitness tracker market, competing alongside other major brands such as Apple, Garmin, and Samsung ([29]). Its consistent product innovation, accessibility, and affordability across multiple device tiers allow it to cater to a diverse and global consumer base. Given its strong market presence and continued influence on the wearable health technology industry, Fitbit serves as an ideal case study for analyzing consumer behavior and satisfaction with fitness wearables.

The Google Play Store was chosen as the primary data source for this study due to its large and diverse Android user base, which accounts for approximately 71% of the global smartphone market according to the Google Store press in 2023 and 2024. As one of the largest digital distribution platforms for mobile applications, the Google Play Store hosts a vast collection of user-generated reviews, offering a rich repository of consumer experiences, feedback, and satisfaction indicators. Unlike traditional survey methods, which may suffer from self-selection bias or limited respondent pools, online review platforms provide naturally occurring consumer insights, reflecting both spontaneous user reactions and long-term product experiences. This dataset provides a comprehensive perspective on real-world user interactions with Fitbit devices, allowing for a qualitative examination of consumer sentiment, perceived benefits, and potential areas of improvement.

### 3.2. Study Setting and Data Collection

The collection period spanned from March to June 2021, a strategic timeframe selected to capture user reactions to significant technical updates introduced in Fitbit’s new product, Charge 4, and activity level during this period (see Figure 1). The updates of Charge 4 included enhanced sleep tracking algorithms, improved heart rate monitoring accuracy, and expanded third-party app integrations, which significantly influenced user experiences and perceptions according to the Fitbit Community. Given the importance of understanding how users respond to major technological advancements, this timeframe provided a unique opportunity to analyze consumer satisfaction following a new product enhancement phase. While newer Fitbit models and software updates have since been released, the fundamental factors driving consumer satisfaction—such as usability, tracking accuracy, and feature integration—remain highly relevant to ongoing research on fitness wearables.

During this period, 38,803 consumer reviews were extracted from the Google Play Store, one of the largest platforms hosting Fitbit application reviews. Google Play was chosen as the primary data source due to its extensive and diverse user base, which includes a wide range of consumers using Fitbit devices across different demographic segments. To collect these reviews, an automated web crawling technique was employed using Python-based tools such as Selenium 4.28.1. and BeautifulSoup 4, ensuring an efficient and comprehensive gathering of user feedback within the specified timeframe. The data extraction process followed a structured methodology, including duplicate removal to eliminate redundant reviews, text normalization to correct spelling inconsistencies, and language filtering to ensure consistency in analysis by focusing exclusively on English-language reviews. Additionally, metadata such as timestamps and user ratings were retained for further contextual analysis. To enhance transparency and replicability, this web crawling process adhered to a systematic and reproducible approach, ensuring the collection of an unbiased dataset that represents real consumer opinions.

This study followed strict ethical guidelines, ensuring that no personally identifiable information was collected, stored, or analyzed. Only aggregate trends and general sentiments were examined to protect user anonymity, and the research methodology adhered to the ethical standards of publicly available online data collection. This study’s emphasis on aggregate consumer sentiment rather than individual user profiling ensures compliance with data privacy standards and maintains the integrity of research ethics.

This study intentionally focused on 5-star reviews to isolate positive experiences and identify key drivers of user satisfaction. This approach provides insights into which features and functionalities resonate most with users following a major update. However, we recognize that this focus introduces positivity bias, as it excludes neutral and negative experiences that may provide critical feedback on product limitations. Prior research has highlighted the tendency for online review patterns to skew toward extreme opinions, meaning that users who have either highly positive or highly negative experiences are more likely to leave reviews ([12]). While this study provides valuable insights into consumer satisfaction, future research should incorporate a more balanced dataset by including reviews with lower ratings. By acknowledging these considerations, this study remains valuable as an exploratory analysis of positive consumer experiences while providing a foundation for future research to expand upon broader consumer perceptions of wearable fitness technology.

### 3.3. Word Frequency Analysis and Word Cloud Analysis

The objectives of word frequency and word cloud analyses were to identify the most frequently used words in consumer reviews about the Fitbit mobile application and smartwatch, as well as to visualize these words in a word cloud. These analyses aimed to uncover the key themes and sentiments in the user feedback.

The collected data were loaded into a pandas Data Frame, a Python library renowned for its data manipulation and analysis capabilities, particularly with tabular data. This step was essential for efficient handling and analysis of the review data. The author commenced word frequency analysis by merging all reviews into a single text string. Using Python’s Counter class from the collections module, a word frequency analysis was conducted to identify the most commonly used words in the dataset. To visually represent these findings, a word cloud was generated with the Word Cloud library in Python, displaying words in sizes proportional to their frequency in the reviews. To enhance these analyses, the author compiled a list of common words for exclusion and various pronouns and prepositions. After reprocessing the reviews to omit these words, a second round of word frequency analysis was performed. This provided a more targeted view of the key terms used in the reviews.

The initial plan to focus on nouns and verbs was hindered by limitations in the available NLP resources. Consequently, the author adopted a simpler method, further extending the list of common words to exclude. This approach aimed to isolate more content-specific words, and the final word frequency analysis was conducted on this extensively filtered data. A concluding word cloud was created using the extensively filtered data, offering a visual representation of the most prominent terms in the reviews after extensive filtering. This methodological approach allowed for a comprehensive analysis of word frequencies in the consumer reviews, with visual aids enhancing the understanding of prevalent themes and sentiments regarding Fitbit products.

### 3.4. Thematic Analysis

The objective of the thematic analysis was to identify and explore recurrent themes in consumer reviews of Fitbit products, providing insights into user experiences, perceptions, and attitudes. The initial phase involved a thorough reading of the reviews to understand the depth and breadth of the content. During this process, preliminary ideas were noted, and recurrent patterns were identified, laying the groundwork for more detailed analysis. The next step was systematically coding the reviews. This involved examining each review to identify significant or interesting features. The coding process was both inductive, driven by the data itself, and deductive, guided by existing theories and frameworks. This meticulous approach ensured that all relevant aspects of the data were captured. After coding, the next phase was to collate these codes into potential themes. This step involved grouping related codes together, examining how different codes might combine to form overarching themes. This process was crucial in moving from specific instances to broader insights about the data.

The potential themes identified were then rigorously reviewed and refined. This involved checking if the themes worked in relation to the coded extracts and the entire dataset. Themes were continuously refined to accurately represent the data, ensuring that they were coherent and meaningful. Once the themes were refined, each was clearly defined and named. This stage involved identifying the essence of what each theme captured about the data. A detailed analysis of each theme was developed, providing clarity and depth to the findings. The final stage of the thematic analysis was writing up the findings. This involved weaving together the analytical narrative with vivid examples from the data. The report contextualized the findings within the broader literature on consumer behavior, technology adoption, and user experience, providing a comprehensive interpretation of the themes.

### 3.5. Concept Map

To explore associations among user-generated content related to Fitbit devices, a concept map was constructed based on word co-occurrence in user reviews. Using a Count Vectorizer, the most frequently occurring words were extracted from the dataset. To maintain conciseness, the analysis focused on the top 30 words. A co-occurrence matrix was subsequently computed using cosine similarity, which quantified the strength of associations between pairs of words.

For the concept map construction, a graph was created using Network X 3.4.0. In this graph, nodes represented the most frequently occurring words, and edges denoted significant associations (cosine similarity > 0.1) between these words. Node and edge weights were adjusted to visualize the relative importance of words and the strength of their associations. Finally, visualization of the concept map was performed. A spring layout was applied to optimize visual clarity, ensuring a balanced distribution of nodes and edges. Nodes were colored uniformly for consistency, while edge thickness varied based on the strength of associations. Labels were added to each node to enhance interpretability.

### 3.6. Sentiment Analysis

To analyze the sentiment of Fitbit user reviews, this study employed TextBlob, a Python-based Natural Language Processing (NLP) tool that utilizes a lexicon-based approach for sentiment classification. Each review was preprocessed by transforming it into a string format, adhering to TextBlob’s input requirements. The sentiment polarity score was then computed on a scale ranging from −1 (negative) to +1 (positive) to classify the sentiment into three categories: (1) Positive Sentiment (score > 0)—reviews expressing satisfaction, ease of use, motivation, or positive experiences; (2) Neutral Sentiment (score = 0)—reviews that contain factual statements, mixed opinions, or lack clear emotional indicators; and (3) Negative Sentiment (score < 0)—reviews highlighting dissatisfaction, technical issues, or unmet expectations. Following classification, sentiment distribution was aggregated to quantify overall user sentiment trends. For visual representation, Matplotlib, a widely used Python visualization library, was employed to generate sentiment distribution graphs. These graphical representations enabled a clear and interpretable visualization of consumer sentiment patterns, facilitating an in-depth understanding of how Fitbit’s features influence user perceptions. This analysis strengthens the current study’s ability to accurately capture and interpret consumer sentiment, providing insights into user experiences with Fitbit’s health-tracking technology.

## 4. Results

### 4.1. Word Frequency Analysis and Word Cloud Analysis

The word frequency analysis and word cloud visualization were conducted on consumer reviews of the Fitbit mobile application and smartwatch. This analysis aimed to identify and visualize the most frequently used words in the reviews, providing insights into the prevalent themes and sentiments among users.

The initial frequency analysis revealed the most commonly used words in the reviews. Common words like “to”, “I”, “and”, “my”, “the”, “it”, “love”, “me”, and “app” were predominant. The word cloud generated from this analysis visually highlighted these frequent words, with their size corresponding to their frequency in the text. This visualization initially pointed to a mix of generic and specific terms related to personal experience and product features. To gain more focused insights, a revised analysis was performed after excluding the above words. This filtering aimed to highlight more content-specific words in the reviews. The revised frequency analysis showed words like “track”, “keep”, “use”, “easy”, “sleep”, and “app” as more prominent. These words indicated a focus on the functional aspects of the Fitbit products, such as tracking capabilities, ease of use, and sleep monitoring. A further refined analysis was conducted, excluding an extended list of common words to isolate more specific terms. The final frequency analysis highlighted words such as “track”, “sleep”, “use”, “easy”, “keep”, “great”, and others, pointing towards specific functionalities and user experiences. Figure 2 indicates user satisfaction rating for each Fitbit feature.

This indicated that users frequently discussed the tracking and monitoring features, appreciated the ease of use, and had positive experiences with the product. The final word cloud, based on the extensively filtered data, provided a clear visual representation of the specific terms that were frequently mentioned in the reviews (see Figure 3). The visualization underscored the importance of tracking features, ease of use, sleep monitoring, and overall satisfaction with the product, as indicated by the size and prominence of these words in the cloud.

The word frequency analysis, particularly after filtering out common words, revealed a significant focus on the functional aspects of the Fitbit products, such as tracking and monitoring capabilities.

The prominence of words like “easy” and “great” in the analysis suggested that users generally found the product user-friendly and were satisfied with their experience. The word frequency analysis and word cloud visualizations provided valuable insights into user perceptions and experiences with Fitbit products. The results highlighted the significance of specific product features and overall user satisfaction, offering a clear perspective on what users value and talk about the most in their reviews.

### 4.2. Thematic Analysis

The thematic analysis of the Fitbit reviews revealed several key themes that provide a comprehensive understanding of the users’ experiences and perceptions. These themes encompass various aspects of user interaction with Fitbit products, from their functionality and ease of use to their impact on users’ lifestyles.

Functionality and Features: This theme revolves around the technical capabilities and specific features of Fitbit devices. In the realm of health technology and consumer electronics, the functionality of a device is crucial in determining its utility and effectiveness. The accuracy of health metrics, the variety of activities tracked, and the integration of these features into a seamless user experience are central to user satisfaction. From a theoretical standpoint, prior research suggests that perceived usefulness is a key determinant of technology adoption, as proposed by TAM ([1]; [7]). While our study did not directly measure perceived usefulness or functionality, the qualitative insights from Fitbit user reviews indicate that features enhancing usability and health monitoring could contribute to users’ adoption decisions ([4]; [6]). Practically, this theme emphasizes the importance of ongoing research and development in health technology. Fitbit can leverage user feedback to innovate and improve features, ensuring that the product remains competitive and meets evolving user needs.

Ease of Use and Interface: This theme highlights the importance of user interface design and ease of use in consumer technology. Prior research in digital consumer behavior suggests that intuitive and user-friendly interfaces enhance engagement and satisfaction in technology acceptance (e.g., [27]). While our study does not quantitatively measure these factors, user reviews frequently highlight the importance of ease of navigation, seamless synchronization, and responsive design as key elements contributing to positive experiences with Fitbit devices. This includes the ease of navigating through the app, readability, and the simplicity of syncing data between the device and the app. This theme is grounded in the concept of usability, a key factor in Human–Computer Interaction (HCI). The usability of a product affects user satisfaction and efficiency, as posited in HCI theories. For Fitbit, focusing on the user interface design is crucial. It involves employing user-centered design principles, conducting usability testing, and continuously refining the interface based on user feedback to improve the overall user experience.

Health and Fitness Motivation: This theme explores the role of Fitbit in motivating users towards a healthier lifestyle. It encompasses how the device’s features like goal setting, progress tracking, and reminders encourage physical activity and health monitoring. The social features, such as sharing achievements or competing with friends, also contribute to this motivation. The theme connects with behavioral theories such as the Self-Determination Theory, which emphasizes the role of intrinsic and extrinsic motivation in adopting health behaviors. Fitbit can act as an extrinsic motivator that gradually fosters intrinsic motivation for health and fitness. Understanding this theme can guide Fitbit in enhancing motivational features, like personalized goal setting, social connectivity, and reward systems, to further engage and motivate users.

Customer Satisfaction and Loyalty: This theme deals with users’ overall satisfaction with the Fitbit product and their loyalty to the brand. Satisfaction encompasses not just the product’s features, but also customer service, value for money, and the brand’s overall image. While our study does not directly measure brand loyalty, prior studies indicate that perceived satisfaction with wearable health technologies contributes to continued usage and consumer trust ([15]; [25]). In our qualitative analysis, Fitbit users frequently referenced long-term engagement and sustained product usage, aligning with existing literature on technology-driven brand loyalty. For Fitbit, this means prioritizing customer feedback, maintaining high-quality standards, and engaging in effective marketing and customer service strategies to foster brand loyalty and positive word-of-mouth.

Personal Impact and Lifestyle Integration: This theme reflects how Fitbit integrates into and influences users’ daily routines and lifestyles. It is not just about tracking fitness but about how the product fits into and enhances various aspects of the user’s life, including sleep, work–life balance, and social interactions. Research on technology acceptance and consumer behavior suggests that products that seamlessly integrate into users’ daily routines foster stronger attachment and continued engagement (e.g., [7]). Fitbit’s ability to provide habit-forming health tracking features aligns with this principle, as many user reviews indicate that the device has become an essential part of their fitness and wellness routines. However, further empirical research is needed to validate the long-term impact of lifestyle integration on brand loyalty and market competitiveness. For Fitbit, recognizing and enhancing this aspect can involve tailoring features to diverse lifestyles, promoting the device’s role in holistic well-being, and ensuring the product’s adaptability to different user needs and routines.

In conclusion, these expanded themes offer a deeper understanding of the various dimensions of user experience with Fitbit devices. Theoretical insights provide a framework for understanding user behavior, while practical implications suggest actionable strategies for product development, marketing, and customer engagement. Understanding these themes can help Fitbit to better cater to its current users and attract new ones by addressing their specific needs and preferences.

### 4.3. Concept Map

The concept map analysis revealed key thematic clusters and associations among words frequently used in Fitbit user reviews, offering valuable insights into user priorities and concerns (see Figure 4). Core themes were identified, with fitness tracking emerging as a prominent cluster. Words such as “track”, “steps”, “calories”, and “easy” were densely connected, indicating a strong user focus on fitness monitoring and the importance of simplicity in tracking features. Additionally, user interface and usability formed another significant theme, as terms like “app”, “sync”, and “friendly” were closely associated. This highlights the importance of seamless interaction and accessibility in the app experience. Health monitoring also featured prominently, with words like “sleep”, “heart”, and “monitoring” clustering together, reflecting the value users place on health-related functionalities.

The analysis also revealed the strength of associations between words. Strong connections between functional terms like “track” and “steps” demonstrated high user engagement with fitness-related metrics. Moderate connections between “battery” and “long” suggested user appreciation for device longevity or a focus on battery-related concerns. However, weak or sparse connections between terms such as “problem” and key functional words like “app” pointed to isolated areas requiring improvement, particularly addressing technical issues in syncing or app navigation.

In practice, Fitbit has product offerings, customer engagement strategies, and overall user experience. The strong clustering of health-related terms such as “sleep” and “heart” indicates user interest in advanced health metrics. Fitbit could introduce features like REM sleep analysis or continuous heart rate monitoring, coupled with actionable alerts for conditions like irregular heartbeat. Additionally, associations between “battery” and “long” highlight expectations for extended battery life, which could be addressed by introducing energy-efficient components or customizable power-saving modes.

The association between terms like “sync”, “app”, and “friendly” underscores the importance of a seamless user interface. Simplifying app navigation through customizable dashboards, where users prioritize specific metrics like steps or heart rate, would enhance usability. Further, embedding a troubleshooting assistant for real-time syncing issues within the app could address user pain points effectively.

Opportunities also exist in personalization and gamification. Fitbit could develop AI-driven personalized recommendations, suggesting tailored activities or goals based on historical user data. Additionally, gamified features like community challenges; leaderboards; or achievement badges for milestones, such as “1 million steps”, would align with user engagement priorities.

Marketing strategies can leverage positive associations like “track” and “easy”. Campaigns could emphasize the simplicity and effectiveness of Fitbit in achieving fitness goals, complemented by testimonials or success stories to build emotional connections with potential customers. Highlighting these aspects could resonate with diverse user demographics.

Weak connections between “problem” and functional terms suggest the need to enhance customer support. Fitbit could introduce dedicated support channels, such as live chat, for resolving common syncing or app issues promptly. Incorporating real-time feedback loops within the app, allowing users to report problems or suggest new features, would demonstrate responsiveness and foster trust.

Strategic partnerships could further broaden Fitbit’s impact. Collaborations with healthcare providers could enable integrated health monitoring solutions, allowing user-approved data to inform preventive care or chronic condition management. Partnering with fitness influencers or communities to promote Fitbit’s motivational benefits through social challenges could also strengthen brand presence and user engagement.

### 4.4. Sentiment Analysis

The sentiment analysis of consumer reviews about the Fitbit mobile application and smartwatch use experiences yielded insightful results. The analysis was based on a dataset comprising various user reviews, each expressing opinions and experiences related to Fitbit products.

Positive Reviews (a sentiment score of 0.318): The analysis revealed that a significant majority, approximately 85.2%, of the reviews were positive. These reviews likely encompassed praise for the product’s features, it’s usability, and the perceived benefits of using Fitbit devices. Prior studies indicate that positive consumer sentiment in user-generated reviews is often linked to satisfaction with a product’s functionality and usability (e.g., [15]). In our analysis of Fitbit user reviews, frequent expressions of appreciation for tracking accuracy, ease of use, and health monitoring features suggest a generally positive reception of the product.

Neutral Reviews (a sentiment score of 0.0): Around 12.9% of the reviews were classified as neutral. Neutral reviews typically include factual statements, mixed feelings, or ambiguities in the expression of clear sentiment. They may provide valuable insights into the aspects of the product that neither particularly please nor disappoint users.

Negative Reviews: A small fraction, about 1.9%, of the reviews were negative. These reviews are crucial for identifying areas of potential improvement. Negative sentiments often highlight specific issues or dissatisfactions that users encounter, such as technical problems, design flaws, or unmet expectations regarding the product. A detailed thematic analysis revealed three primary areas of dissatisfaction as understanding these dissatisfaction areas is critical for enhancing product design and user experience: (1) synchronization issues (e.g., “Syncing is unreliable; I sometimes lose my step data after an update” and “The app often fails to sync with my phone, making tracking inconsistent”)—these reviews indicate that data loss and synchronization errors negatively impact user trust in Fitbit’s reliability; (2) battery life concerns (e.g., “Battery drains quickly, even when notifications are turned off” and “I have to charge my Fitbit every two days, which is frustrating”)—while Fitbit offers varying battery life across models, the dissatisfaction suggests a need for optimized power management settings; and (3) app performance/bugs (e.g., “The app crashes sometimes when I try to check my stats” and “After the last update, my step count is not displayed correctly”)—these reviews highlight concerns regarding software updates negatively affecting functionality, emphasizing the importance of app stability in user retention. To address these challenges, Fitbit should prioritize real-time error detection for syncing failures, improve cross-device compatibility, and enhance software testing protocols before updates. Transparency regarding battery consumption across different usage scenarios can also help manage expectations. Stability should take precedence over frequent updates, and an in-app feedback mechanism would allow rapid bug detection and resolution. Strengthening trust in data accuracy and device efficiency will be crucial for sustaining engagement.

The overwhelming predominance of positive reviews suggests a high level of user satisfaction with Fitbit’s mobile app and smartwatch. This is indicative of successful product design and functionality that aligns well with user needs and expectations. The minimal presence of negative reviews points to a generally favorable reception of the product but also serves as a reminder of the importance of addressing specific user concerns to maintain and enhance product quality and user experience (see Figure 5). The sentiment analysis results paint a largely positive picture of the consumer experience with Fitbit’s mobile application and smartwatch. The data underscore the success of the product in meeting most user expectations, while also highlighting the importance of continual improvements and responsiveness to all types of user feedback. This analysis provides valuable insights for product development, marketing strategies, and customer service approaches.

## 5. Conclusions and Implications

### 5.1. Theoretical Implications

TAM has been a cornerstone in understanding how users come to accept and use technology ([2]; [6]; [25]). The empirical findings from the Fitbit reviews offer a nuanced expansion of this model, particularly in the realm of health technologies. Although our study did not directly measure perceived ease of use or perceived usefulness, existing literature suggests that these factors are fundamental to technology adoption in health and fitness contexts ([35]). User feedback from our qualitative analysis highlights the importance of intuitive interfaces, real-time tracking, and usability features, which are theoretically linked to the TAM framework ([4]; [6]). However, the Fitbit context introduces a unique dimension to TAM: the integration of health consciousness into technology acceptance ([14]; [15]). This suggests that in the health technology domain, acceptance is not merely a function of usability or utility but is also deeply intertwined with personal health goals and awareness ([25]; [35]). Therefore, this study enriches the TAM framework by incorporating health-specific motivations into the model, offering a more comprehensive understanding of technology acceptance in health and fitness contexts ([15]).

Fitbit’s role in fostering intrinsic motivation for health and fitness aligns with and expands upon the principles of Self-Determination Theory ([10]; [13]; [24]). Self-Determination Theory posits that intrinsic motivation is more sustainable and effective for behavior change than extrinsic motivation ([22]; [30]). The user reviews revealed how Fitbit’s features like goal setting, feedback, and social sharing can initially act as extrinsic motivators. Over time, however, these features can help users internalize their motivation for physical activity, shifting from extrinsic to intrinsic motivation ([10]; [22]). Specifically, users often start engaging with Fitbit due to external incentives such as badges, challenges, and step goals, which drive initial motivation (e.g., “At first, I used Fitbit to win step challenges with friends. Now, I don’t need challenges—I just love hitting my personal bests!” and “My Fitbit kept me motivated at first with badges, but now I just enjoy seeing my progress every day”). These testimonials indicate that Fitbit’s gamified mechanics serve as external reinforcements, aligning with Self-Determination Theory’s proposition that extrinsic motivators can initiate engagement in a behavior before it is internalized ([30]). As users continue engaging with Fitbit, their motivation shifts from external rewards to intrinsic engagement, where activity tracking becomes a natural, self-sustaining habit. The following user reviews exemplify this transition (e.g., “Since having my first Fitbit in 2017, I haven’t missed a day of tracking—it’s second nature now” and “Tracking my steps became a daily habit—I don’t think about it anymore; it’s just part of my routine”). This suggests that frequent reinforcement of behaviors—particularly through real-time feedback—gradually turns them into automatic, habitual actions. Finally, beyond habit formation, Fitbit contributes to self-perception as an active individual, reinforcing long-term engagement through identity alignment (e.g., “My Fitbit is part of me—I feel incomplete without it” and “I love my Fitbit! It keeps me motivated and reminds me that I am an active person”). This progression posits that individuals are more likely to sustain behaviors when they align with their self-concept and personal values ([10]).

This finding is significant as it provides empirical evidence supporting the Self-Determination Theory in the context of wearable health technologies ([13]; [22]; [24]). It suggests that fitness wearables can be instrumental in transitioning users’ motivation from being externally driven to self-driven, thereby having a more lasting impact on their health behaviors.

The findings also have implications for broader health behavior theories. By demonstrating how a technological device like Fitbit can influence users’ physical activities and health routines, this study contributes to an understanding of behavioral change in the digital age. It suggests that wearable technologies can be powerful agents in modifying health behaviors, aligning with theories that emphasize the role of environmental and contextual factors in health behavior change ([12]; [19]; [34]). This extension of health behavior theories to incorporate technological influences reflects the evolving landscape of health behavior determinants in the modern world, where technology increasingly acts as a pivotal factor ([34]).

Lastly, this research provides valuable insights into user-centered design theory. User-centered design posits that the success of a product hinges on how well it meets the needs and expectations of its users ([8]; [20]). The analysis of Fitbit reviews illustrates this principle in action, showing that user satisfaction is closely linked to how well the product features align with user needs, particularly in terms of functionality, ease of use, and integration into daily life. This finding underscores the importance of continuous user feedback and iterative design processes in developing health technology products ([8]). It suggests that a deep understanding of user experiences and preferences should be at the heart of health technology design, reinforcing the principles of user-centered design theory ([8]; [20]).

In conclusion, the detailed analysis of Fitbit reviews offers substantial contributions to theoretical frameworks such as TAM, Self-Determination Theory, health behavior theories, and user-centered design theory. By providing empirical evidence from the context of wearable health technologies, this research enriches these theoretical frameworks, offering new insights and understandings that are crucial in the evolving landscape of health technology and behavior.

### 5.2. Managerial Implications

From a practical perspective, the demand for accuracy in health metrics highlights the need for advanced technological integration in Fitbit products. It is not just about adding new features; it is about enhancing the precision and reliability of existing health-tracking capabilities. This involves leveraging cutting-edge biometric sensing technologies and sophisticated data analytics to ensure that users receive actionable and meaningful health insights rather than raw data points. Specifically, Fitbit could focus on integrating advanced machine learning algorithms that provide predictive health alerts and personalized fitness recommendations. For example, if a user’s resting heart rate is consistently elevated over a period of time, the app could generate a prompt stating “Your heart rate has been trending higher this week. Consider incorporating stress-relief activities or monitoring hydration levels”. Similarly, for sleep tracking, rather than just reporting on sleep duration, Fitbit could offer contextual guidance, such as “Your deep sleep has decreased by 15% over the past week. Try limiting screen exposure before bedtime to improve sleep quality”. Such AI-driven, real-time feedback would reinforce Fitbit’s role as an active health coach rather than a passive tracking device, enhancing its perceived usefulness and long-term engagement.

Beyond data accuracy, the significance of an intuitive user interface extends beyond basic usability. It encompasses creating an experience that is both accessible to a diverse user base and adaptable to individual preferences and behaviors. Many users in this study reported frustration with syncing failures, app crashes, and a lack of personalized interactions. To address this, Fitbit could develop an adaptive user interface that learns from user interactions, dynamically adjusting to offer a more personalized experience. For instance, if a user frequently checks their step count and heart rate but rarely engages with calorie tracking, the app could prioritize displaying these preferred metrics on the dashboard, reducing clutter and improving user experience. Additionally, broader accessibility features, such as voice-command functionalities for hands-free navigation and enhanced visual contrasts for users with visual impairments, would make the devices more inclusive and user-friendly across diverse demographics.

The motivational aspect of Fitbit devices offers a unique angle for emotionally resonant marketing. However, empirical findings suggest that static gamification features, such as step challenges and achievement badges, lose their appeal over time. To sustain user engagement, Fitbit should move beyond traditional incentives and integrate competence-enhancing gamification mechanics that reinforce long-term motivation. Adaptive goal-setting algorithms should dynamically adjust fitness targets based on past behavior, ensuring that users remain challenged without feeling discouraged. For example, users who consistently exceed their step goal could receive incremental challenges, while those struggling to meet their targets could be encouraged with smaller, attainable milestones, such as “Let’s aim for 7500 steps today—small steps lead to big changes!” Furthermore, progress-based leveling systems could be introduced, where users “unlock” new guided workouts or personalized AI-generated fitness recommendations based on their activity trends. Fitbit could also revamp its social challenge system by enabling users to create customized challenges (e.g., “7-day mindfulness walking challenge” or “Weekend step competition with friends”), fostering a stronger sense of community ownership. Since user engagement is deeply tied to social accountability, these enhancements would not only sustain motivation but also encourage habit formation and long-term adherence.

Integrating customer feedback should move beyond passive collection to active engagement and co-creation with users. This approach not only helps in refining products but also builds user loyalty and trust. Establishing a user advisory panel for continuous feedback and co-creation of features could be a strategic move, allowing Fitbit to iteratively refine its functionalities based on real user needs. Additionally, incorporating in-app feedback loops where users can report syncing issues or suggest new features in real time would enhance responsiveness and customer satisfaction. Our analysis also highlighted user frustration with technical reliability, particularly in synchronization errors and battery life concerns. To address this, Fitbit should implement predictive error detection that notifies users of potential syncing failures before they occur, with suggestions like “We noticed a delay in syncing. Try refreshing Bluetooth or restarting your device”. Further, improving offline functionality, such as ensuring that activity tracking remains accurate even without internet connectivity, would prevent frustration and enhance the overall user experience.

Fitbit’s role in broader health promotion initiatives should extend beyond fitness tracking into holistic wellness domains, aligning with its growing use as a preventive health tool. Strategic partnerships with medical research institutions could enable Fitbit to refine its health monitoring features and potentially contribute to early detection of health conditions such as irregular heart rhythms or sleep disorders. Collaborating with mental health platforms to study the correlation between physical activity and mental well-being could lead to the integration of mindfulness-based features, such as guided meditation prompts when stress levels appear elevated. Additionally, Fitbit could explore corporate wellness partnerships, integrating its devices into employee health programs, where companies incentivize participation in step challenges and wellness initiatives. Expanding into the educational sector, such as collaborating with schools to integrate Fitbit technology into physical education programs, could provide data-driven insights into student activity levels and encourage lifelong healthy habits.

App developers aiming to maximize customer satisfaction should begin by systematically analyzing user feedback (see Figure 6). This process involves collecting and organizing feedback from diverse sources, such as app store reviews, customer surveys, support tickets, and social media discussions. Employing Natural Language Processing (NLP) tools can streamline the analysis of large datasets, identifying recurring themes and sentiments efficiently.

After collecting feedback, the next step is to categorize and prioritize the issues. Grouping feedback into themes, such as health features, usability, or technical glitches, allows for a structured approach to addressing user concerns. Issues should then be prioritized based on their frequency, severity, and impact on user satisfaction. For example, technical issues like syncing problems may take precedence over minor feature requests. Validation of these issues using internal analytics, such as crash logs or usage patterns, ensures that user-reported problems align with observable data.

When addressing health-related issues, developers should enhance the accuracy and insights of health metrics. This includes refining features like sleep and heart rate monitoring to ensure reliability. Providing actionable insights, such as personalized sleep hygiene tips or fitness recommendations, can further engage users. Developers should also focus on accessibility, ensuring that health data are visually intuitive and easy to interpret for all users.

For usability concerns, improvements should focus on simplifying navigation and optimizing performance. Streamlining app workflows to reduce steps for common tasks and introducing interactive onboarding guides can significantly enhance the user experience. Customizable dashboards that allow users to tailor their interface to display preferred metrics are another effective enhancement. Additionally, ensuring compatibility with assistive technologies, such as screen readers, and incorporating larger fonts for visually impaired users can broaden accessibility.

Before rolling out updates, developers should review the effectiveness of their changes. Conducting beta testing with a small user group helps identify any remaining issues and allows for iterative improvements. Monitoring key metrics, such as engagement rates and customer retention, provides quantitative evidence of improvement impact. Collecting qualitative feedback from beta testers further ensures a comprehensive understanding of user perceptions. Regression testing is critical to confirm that new changes have not introduced unintended issues.

Once validated, updates should be deployed with clear communication strategies. Detailed release notes that highlight new features and fixes, combined with marketing campaigns to announce significant updates, can effectively re-engage users. Continuous monitoring post-update ensures that user expectations are met, and that new feedback loops within the app can help developers capture ongoing input.

Finally, fostering a cycle of continuous improvement is essential. Regularly collecting user input on new features or enhancements and building engagement through community-driven features, such as leaderboards or interactive challenges, can maintain user satisfaction. Iterative development based on these insights ensures that the app evolves to meet changing user needs, creating a dynamic and responsive user experience.

In conclusion, these expanded in-depth practical implications provide a detailed framework for how Fitbit and similar companies can enhance their product offerings, marketing strategies, customer engagement, and partnership initiatives. By focusing on these key areas, these companies can not only improve their products and user experiences but also play a pivotal role in advancing public health and personal wellness.

### 5.3. The Impact of Fitbit on Long-Term Health Behaviors

Fitbit’s long-term impact on user behavior extends beyond mere fitness tracking, fostering habit formation, intrinsic motivation, and identity reinforcement ([4]). Many users report that daily tracking becomes an automatic behavior, aligning with habit formation theory, which suggests that consistent reinforcement transforms conscious actions into long-term routines ([21]; [36]). A review from a long-term user illustrates this transition: “*I’ve had Fitbits for over seven years now. I can’t imagine NOT tracking my steps and sleep”.* Initially, Fitbit’s gamification features and goal-setting mechanics serve as extrinsic motivators, encouraging engagement through badges and challenges (“At first, I used Fitbit to win step challenges with friends. Now, I don’t need challenges—I just love hitting my personal bests!”). However, over time, users experience a shift toward intrinsic motivation, engaging in physical activity for personal satisfaction rather than external rewards ([21]; [36]). This aligns with Self-Determination Theory ([28]), which posits that autonomy-supportive tools facilitate self-motivation and sustained engagement. Additionally, Fitbit becomes an integral part of users’ health-conscious identity, as expressed in reviews like “*My Fitbit is my accountability partner—I don’t need reminders anymore; I just use it daily”* and “*It’s more than a fitness tracker now; it’s a reflection of my health journey”.* These testimonials illustrate how wearable technology can reinforce self-perception as an active individual ([4]). To enhance this behavioral transformation, Fitbit could implement AI-driven adaptive goal recommendations, personalized habit-tracking dashboards, and long-term usage milestones that solidify Fitbit’s role as a lifelong health companion rather than just a fitness tracker. These findings suggest that wearable technologies can function as powerful enablers of sustained health behaviors, fostering deep user engagement through habit reinforcement and identity integration ([21]; [36]).

### 5.4. Limitations and Directions for Future Research

The analysis was limited to publicly available user reviews. These reviews may not fully represent the entire spectrum of user experiences, as they tend to capture more extreme sentiments (either very positive or very negative). The reliance on five-star reviews, in particular, introduces positivity bias, as it does not account for neutral or dissatisfied users. This bias limits the ability to assess critical concerns that may affect adoption and long-term engagement with Fitbit devices. Future studies could broaden the data scope by including reviews with lower ratings (e.g., one- to three-star reviews), along with survey responses, interviews, or focus groups to gain a more balanced and comprehensive understanding of user experiences.

In addition, our study predominantly relied on qualitative thematic analysis, which, while insightful, is inherently subjective and may carry biases of interpretation. The qualitative approach limits the ability to establish statistical relationships between key adoption factors. Subsequent research might incorporate more diverse and objective quantitative methods, such as structural equation modeling (SEM) or regression analysis, to empirically test relationships between key variables, such as perceived usefulness, ease of use, and behavioral intention to use fitness wearables.

Furthermore, the reviews analyzed did not explicitly account for cultural and demographic diversity, which can significantly influence user experiences and perceptions. While Fitbit has a global user base, differences in age, gender, cultural background, socioeconomic status, and health conditions may impact technology acceptance and engagement with wearable devices. Thus, future research should consider these demographic factors and employ segmented analyses to explore whether technology acceptance varies across different populations.

A major concern raised by the reviewer relates to the data collection period (March–June 2021). Given the rapid evolution of fitness technology, reliance on a dataset from 2021 may limit the generalizability of findings, as newer Fitbit models and software updates may have introduced improved features or addressed previously reported user concerns. While this study provides valuable insights into Fitbit adoption during this period, future research should analyze longitudinal datasets or conduct follow-up studies incorporating more recent user feedback to assess whether the identified themes remain consistent or have evolved over time.

Finally, the dynamic nature of the health and fitness wearable industry means that study findings are time-sensitive and may become less relevant as technology advances. Ongoing research is necessary to track how new features, software updates, AI-driven health insights, and integrations with other health platforms impact user engagement. Addressing these limitations and pursuing the suggested directions for future research would not only strengthen the understanding of fitness technology’s role in health and wellness but also guide future innovations, policy recommendations, and interventions in this rapidly evolving field.

## Figures and Tables

**Figure 1 behavsci-15-00231-f001:**
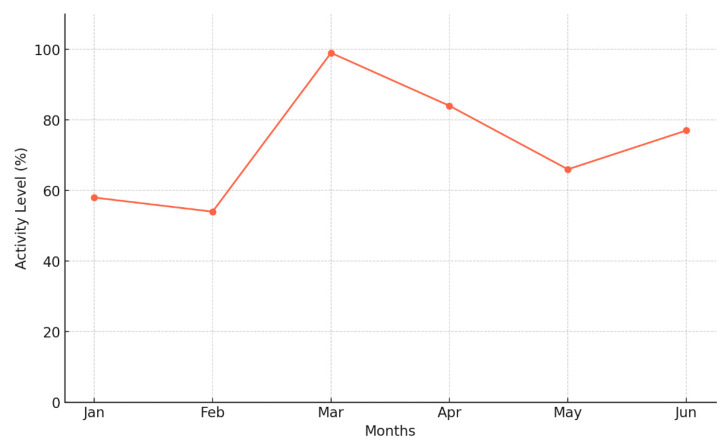
Increase in user physical activity over time.

**Figure 2 behavsci-15-00231-f002:**
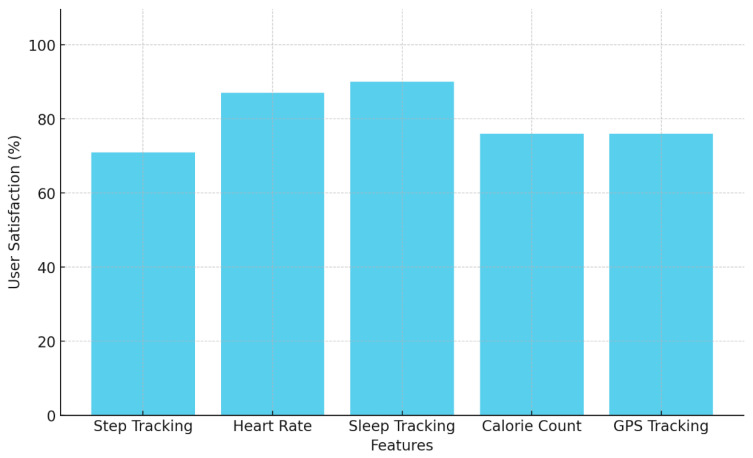
User satisfaction rating for Fitbit features.

**Figure 3 behavsci-15-00231-f003:**
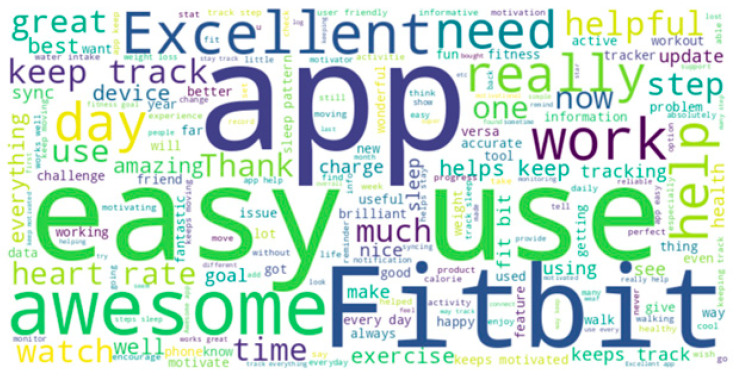
Word cloud of Fitbit reviews (excluding specific words).

**Figure 4 behavsci-15-00231-f004:**
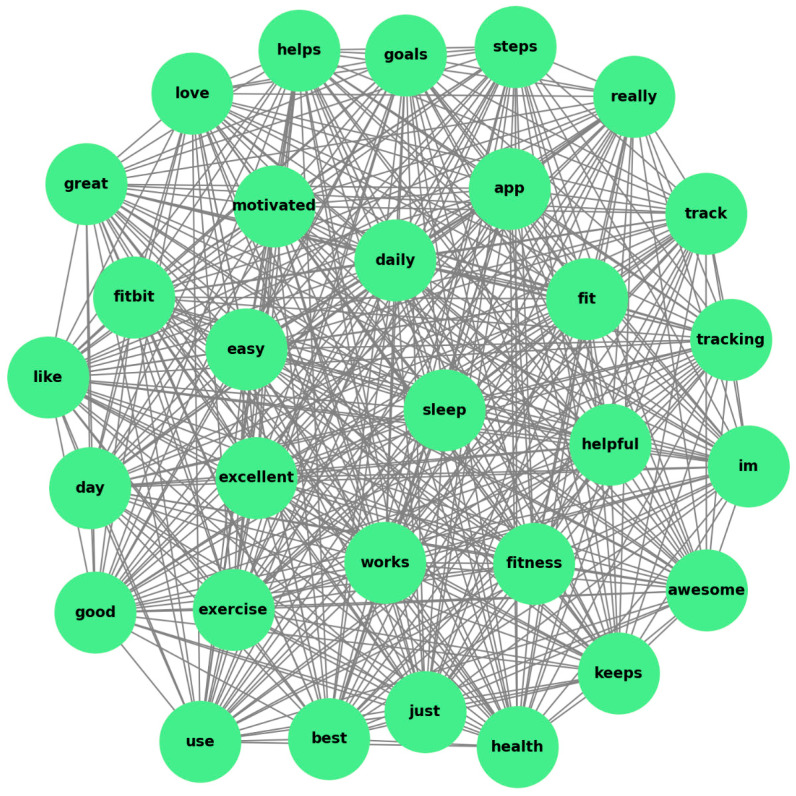
Concept map of word association.

**Figure 5 behavsci-15-00231-f005:**
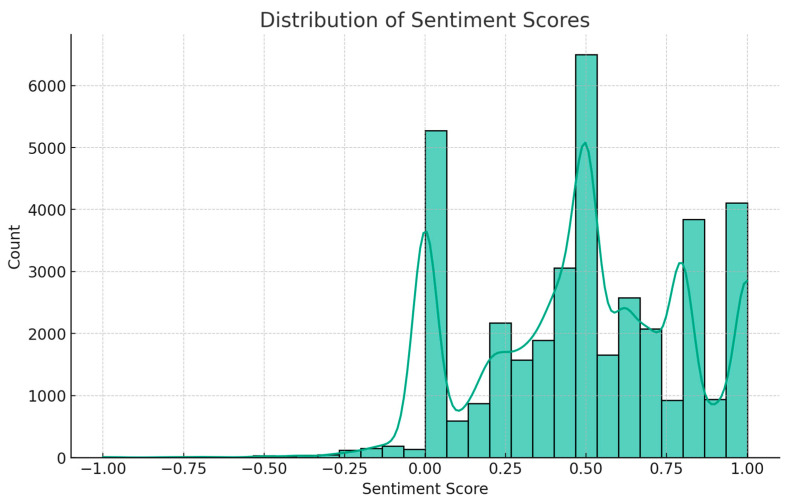
Result of sentiment analysis.

**Figure 6 behavsci-15-00231-f006:**
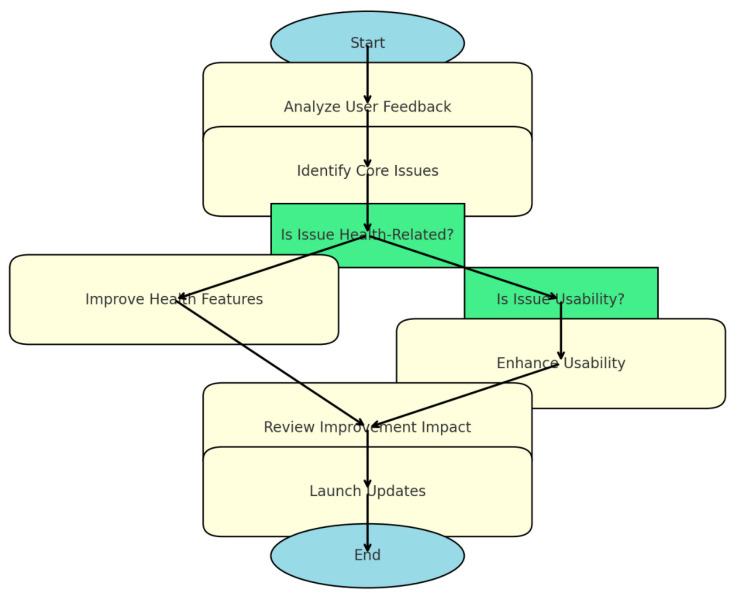
Flowchart for maximizing customer satisfaction in app development.

## Data Availability

The data presented in this study are available on request from the corresponding author. The data are not publicly available due to privacy reasons.

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
