# Peer review of "Digital Fitness Revolution: User Perspectives on Fitbit’s Role in Health Management"

_behavsci, 2025, doi:10.3390/bs15020231_

Round 1

Reviewer 1 Report

Comments and Suggestions for Authors

Congratulations on the work done. It seems to me to have originality. However, I would like to highlight a few comments in case the article could be improved. 

-The section on the Technology Acceptance Model (TAM) is sound, but it would be useful to clarify how ‘health awareness’ was measured in Fitbit uptake.

-In the discussion on Self-Determination Theory (SDT), it is mentioned that Fitbit features help users to move from extrinsic to intrinsic motivation. However, it would be beneficial to include more data or specific examples to support this claim.

-Some ideas are repeated in different sections. Unifying the key points would avoid redundancies.

-Although the results of the sentiment analysis are mentioned, there is no explanation of how the sentiment analysis was conducted. Was a natural language processing (NLP) tool used? Was there a methodology for determining the sentiment score (e.g. a scale from 0 to 1)? It would be important to include details about the tools and the process used to validate the results.

-Breakdown of negative reviews: Although it is mentioned that negative reviews are small, it would be useful to explore in more detail what specific areas generated user dissatisfaction. Was it a problem with product features, or were there recurring issues in the user experience that could be addressed? Drilling down into the types of negative reviews could provide more useful insights for product improvement.

-Limited empirical evidence: Although key theories and concepts are mentioned, the article lacks specific examples of how Fitbit implements these principles. For example, it is mentioned that the device can help users move from extrinsic to intrinsic motivation, but no concrete data or case studies are provided to support this claim. It would be useful to include more direct empirical evidence, such as user examples or longitudinal analyses that demonstrate this shift in motivation.

Reviewer 2 Report

Comments and Suggestions for Authors

The study presents relevant content. However, there are aspects that need significant improvements and additional clarifications. The arguments and discussion of the findings could be more coherent and convincing. The conclusions of the article also need to be better grounded in the results presented or in secondary literature. The following analysis details these observations and suggests possible improvements to strengthen the work.

In the introduction, authors should reinforce why they opted for the use of TAM instead of other more recent models of technological acceptance more directed towards the customer's perspective, such as UTAUT 2.

In the literature review, authors should include more recent references. Technology in fitness is rapidly evolving, and more recent studies may offer additional insights.

Still in the literature review, the authors should add an approach to existing technological adoption models, with reinforcement on the model used. Thus, authors should also provide a better explanation of the TAM model and its dimensions.

Methodologically, on line 47, the authors indicate "Fitbit was selected as the primary focus of this research due to its prominence in the health and fitness technology market." Please include a reference to substantiate this information.

Also, authors should provide more details about the data collection process, such as the web crawling technique used, to increase the transparency and replicability of the study.

My main concern is with the study is the data collection period. It refers to 2021. In a rapidly evolving field like technology, data from 2021 may no longer be relevant. The authors should justify why more recent data was not used, as this raises questions about its validity.

Although, authors indicate that "March to June 2021, a strategic timeframe selected to capture user reactions during the peak seasons following a significant technical update to the Fitbit application and devices," more justifications are needed.

In my opinion, despite the quantity (38,803) of reviews analyzed, this approach to evaluating consumer satisfaction has some limitations. The positivity bias, as focusing on 5-star reviews may ignore constructive criticism that could help improve the product. 5-star reviews may not represent the experience of all users, especially those who had negative or average experiences. Positive reviews can be influenced by external factors, such as marketing campaigns or incentives for positive reviews.

In the results, especially in topics 4.2 and 4.3, statements are made that cannot be supported based on the methodology used and the results obtained in this study. Most statements also lack confirmation through studies. They do not have scientific robustness or foundation, as the authors do not cite any studies to support them. Examples of this are phrases like: "A well-designed interface that is intuitive and user-friendly can significantly enhance user engagement and satisfaction."; "This theme is rooted in consumer satisfaction theory, which posits that satisfaction is a key determinant of brand loyalty and repeat purchasing behavior. A positive user experience leads to higher customer retention and brand advocacy."; "This theme aligns with the concept of lifestyle integration in consumer behavior studies, which suggests that products successful in integrating into users’ lifestyles become indispensable and gain a competitive edge."; "Positive sentiments often reflect customer satisfaction and a favorable reception of the product's features and overall functionality."

In the conclusions and limitations section, on line 306: "From a theoretical standpoint, this theme aligns with the Technology Acceptance Model (TAM), which suggests that perceived usefulness significantly influences technology adoption. Effective functionality enhances perceived usefulness, encouraging continued use and positive attitudes towards the technology" and line 449 "The reviews underscore the importance of perceived ease of use and perceived usefulness – two central tenets of TAM– in determining the acceptance of fitness wearables," I believe these statements cannot be made as these dimensions were not directly measured in the study. If these dimensions were not measured, it would be more appropriate for the authors to mention this relationship as a hypothesis or a suggestion for future research, rather than a direct conclusion based on the study data. They could rephrase the statement to something like: "Although our study did not directly measure perceived usefulness and functionality, existing literature suggests that these factors are important for technology adoption, according to the Technology Acceptance Model (TAM). Future research could explore this relationship in more detail." And please include a reference to substantiate this information.

The limitations section is mentioned but could be expanded to include more details about methodological limitations and how they may have impacted the results.

The practical implications are well discussed but could be more specific. For example, providing concrete examples of how app developers can implement the recommendations could be helpful.

The references are appropriate to the context of the study. Authors follow APA norms. All references cited in the text are present in the reference list. Although not mandatory, the inclusion of DOI is recommended, and the authors do not include it.

Round 2

Reviewer 2 Report

Comments and Suggestions for Authors

All suggestions and comments have been carefully considered and addressed, leading to significant improvements in the manuscript. As a result, my final decision is to accept the article for publication. Congratulations to the authors on their work.